# Association of frequency of television watching with overweight and obesity among women of reproductive age in India: Evidence from a nationally representative study

Rajat Das Gupta[1,2,3]*, Shams Shabab Haider[2], Ipsita Sutradhar[1,2], Mohammad Rashidul Hashan[4], Ibrahim Hossain Sajal[2,5], Mehedi Hasan[1,2], Mohammad Rifat Haider[6], Malabika Sarker[1,2,7]

1 Centre for Non-Communicable Diseases and Nutrition, BRAC James P Grant School of Public Health, BRAC University, 68 Shaheed Tajuddin Ahmed Sarani, Mohakhali, Dhaka, Bangladesh, 2 Centre for Science of Implementation & Scale-Up, BRAC James P Grant School of Public Health, BRAC University, 68 Shaheed Tajuddin Ahmed Sarani, Mohakhali, Dhaka, Bangladesh, 3 Department of Epidemiology and Biostatistics, Arnold School of Public Health, University of South Carolina, Columbia, South Carolina, United States of America, 4 Dhaka Medical College and Hospital, Dhaka, Bangladesh, 5 Department of Mathematical Sciences, School of Natural Sciences & Mathematics, The University of Texas at Dallas, Dallas, Texas, United States of America, 6 Department of Social and Public Health, College of Health Sciences and Professions, Ohio University, Athens, Ohio, United States of America, 7 Institute of Public Health, University of Heidelberg, Heidelberg, Germany

* rajat89.dasgupta@gmail.com

## Abstract

### Background

For women of reproductive age, overweight and obesity are an established risk factor for several medical complications. To address the increasing rate of obesity in India through public health awareness programs, the association between common behaviors and over-weight and obesity needs to be investigated. This study aims to determine whether there is any association between the frequency of television watching and overweight and obesity among women of reproductive age (15–49 years) in India.

### Methods

This is a cross-sectional study that utilized data from the National Family Health Survey (NFHS-4), which utilized a nationally representative sample from all 29 states and 7 union territories of India. The survey itself followed a two-staged stratified random sampling technique. The primary outcome of interest was overweight (23.0 kg/m$^2$ to <27.5 kg/m$^2$) and obesity ($\geq$27.5 kg/m$^2$), measured by using the Asian body mass index cut-off. The major explanatory variable was the frequency of television watching, measured in days per week. Sample weight of NFHS-4 was adjusted during the analysis. Multilevel ordered logistic regression was conducted to identify the factors associated with overweight and obesity. To show the strength of association, both the unadjusted Crude Odds Ratio (COR) and the

**Data Availability Statement:** Data were collected and owned by the demographic and health survey

authority. Data are available at: https://dhsprogram.com/data/dataset/India_Standard-DHS_2015.cfm?flag=0. Following instruction, data are available to download. Anyone interested to work with these data will be able to access these data in the same manner as the authors.

**Funding:** This article received no external funding.

**Competing interests:** The authors have declared that no competing interests exist.

Adjusted Odds Ratio (AOR) were reported with a 95% confidence interval (CI). A *p*-value<0.05 was considered statistically significant.

## Results

The analysis included weighted data from 644,006 Indian women of reproductive age (15–49 years). Among the respondents, 33.5% were overweight or obese (BMI $\geq$23.0 kg/m$^2$). The prevalence of overweight and obesity increased with age (*p*-value <0.0001) and almost half of the women aged 35–49 years were either overweight or obese (48.6%). The prevalence was significantly higher among those living in an urban area compared to a rural area (urban 46.5% vs. rural 26.5%; *p*-value <0.001). The prevalence of overweight and obesity increased with the frequency of watching television and was the highest among the individuals who reported watching television almost every day (*p*-value <0.0001). Women watching television almost every day had 24% (AOR: 1.24, 95% CI: 1.21–1.26; *p*-value <0.001) increased odds of being overweight and obese compared to their counterparts who never watched television.

## Conclusions

This study found that the likelihood of being overweight and obese significantly increased with the frequency of watching television; likely due to physical inactivity during leisure time. Further studies should examine the physical activity and food habits of this target group. Public health promotion programs in India should raise awareness regarding the harmful effects of the sedentary lifestyle associated with watching television.

## Background

Overweight and obesity was the fourth leading cause of global mortality in 2017, as per the estimate published by the Global Burden of Disease (GBD) [1]. This condition is attributable to several non-communicable diseases (NCDs), including cardiovascular diseases, Type 2 diabetes mellitus, cancer and chronic kidney disease [2–4]. Overweight and obesity has emerged as a leading public health problem as the prevalence of this condition is increasing day-by-day at an alarming rate [5]. Between 1975 and 2016, the global prevalence of obesity among women increased 465% (from 69 million in 1975 to 390 million in 2016) [6]. This global obesity epidemic has also affected the South and Southeast Asian countries [7]. India is currently undergoing a nutritional transition with an increased burden from overweight and obesity, especially among women, which in turn, also predisposes them to a variety of NCDs [8–10].

One of the main drivers of the global obesity epidemic is the transition to an urban lifestyle. Urban, as opposed to rural, environments have facilitated an improvement of the transportation system. Adequate infrastructure allows residents of cities to access the booming fast food industry. The increased availability of junk foods, as well as the reduced space for physical activities are a consequence of this rapid transition to urban living [11–13]. Another key impact on the epidemic is the increased adoption of a sedentary lifestyle, which has been observed to closely relate to time spent watching television as a leisure activity. The reduced energy expenditure of this leisure activity is closely associated with overweight and obesity [14,15]. Other possible mechanisms include frequent exposure to food and beverage advertisements and the subsequent intake of those foods and beverages; sleep disturbance and reduced

involvement in outdoor physical activity [16–18]. Women of reproductive age that settle into a sedentary lifestyle are at risk of being overweight/obese [19]. For women of reproductive age, overweight and obesity is an established risk factor for several medical complications, including pre-eclampsia, eclampsia and gestational diabetes mellitus (GDM) [20,21]. Overweight and obese women having GDM also tend to develop diabetes mellitus during the post-partum period [22]. Overweight and obesity is attributable to an increased risk of infertility and miscarriage [23]. Living a sedentary lifestyle is also an independent risk factor for infertility in women [24].

In the USA and Australia, increased frequency of television viewing was found to be directly associated with overweight and obesity [25,26]. A similar association was also revealed in the case of women of reproductive age in Bangladesh and Myanmar [15,27]. However, this exact association has yet to be explored in the context of India. The objective of this study was to find the association between the frequency of television watching and overweight and obesity among Indian women of reproductive age (15–49 years) using data from the nationally representative National Family Health Survey 2015–16 (NFHS-4).

## Methods

### Data source

NFHS-4 is a survey in India that utilized a nationally representative sample and was implemented between January 2015 and December 2016. The survey was funded by USAID, DFID, the Bill & Melinda Gates Foundation, UNICEF, UNFPA, the MacArthur Foundation and the Government of India. It was implemented by the International Institute for Population Sciences (IIPS) under the patronization of the Indian Ministry of Health and Family Welfare (MoHFW) and with the technical support of ICF International. The objective of this survey was to generate nationally representative data on various indicators related to maternal, neonatal and child health (MNCH) as well as on other emerging issues like NCDs. A more detailed discussion on the methods used for the survey has been published previously [28].

### Study population and survey design

Indian women aged 15–49 years were the study population. NFHS-4 followed a two-staged stratified random sampling. The survey utilized the sampling frame from the 15th Indian Census 2011. Census Enumeration Blocks (CEBs) and villages were held as primary sampling units (PSUs). After selecting the PSUs by the probability proportional to size (PPS) technique, a complete household listing was conducted. The PSUs with at least 300 households were divided into a segment of about 100–150 households. Using probability proportional to segment size, two of the segments were randomly selected. This segment, which is either a PSU or a segment of the PSU, is called cluster. During the second phase, a fixed number of 22 households were selected using a systematic sampling technique from each cluster [28].

### Data collection and measurements

Trained staff carried out the data collection from 20 January 2015 to 4 December 2016. For collecting data related to socio-demographic characteristics, NFHS-4 used a standard women's questionnaire, which was pre-tested, modified and adapted according to the Indian context. Seca 874 digital scales and Seca 213 stadiometers were used to take anthropometrical measurements of weight and height, respectively. The instruments were calibrated before each recording [28].

## Outcome variable and covariates

The main outcome variable of this study was body mass index (BMI), which was obtained by the division of the weight by the height squared of each respondent and expressed in kilogram/meter$^2$ (kg/m$^2$) [29]. Respondents were then divided into three categories based on BMI: (1) Normal or underweight (BMI <23.0 kg/m$^2$), (2) overweight (BMI between 23.0 kg/m$^2$ and <27.5 kg/m$^2$) and (3) obesity (BMI ≥27.5 kg/m$^2$). The ranges were defined based on an Asian-specific BMI cut-off value [30]. The main explanatory variable was frequency of watching television, which was categorized into (1) not viewing television at all, (2) viewing television less than once a week, (3) viewing television at least once a week and (4) viewing television almost every day [28]. Based on the literature, other considered covariates were age group, place of residence, state of residence, marital status, highest educational attainment, wealth quintile, parity and number of household members in the family. Age group was divided into 15–24 years, 25–34 years and 35–49 years. Place of residence was categorized into urban or rural, state of residence was divided into 29 states and 7 union territories. Marital status was divided into single, currently married or separated/divorced/widowed. Highest educational attainment was divided into four categories: no education, primary education, secondary or higher education. Parity was divided into five categories: 0, 1, 2, 3 and >3. The number of household members was divided into a binary selection around the median: ≤5 or >5. The NFHS-4 collected data on selected assets of the households, including household construction materials (e.g. roof and flooring construction material), water source and sanitation facilities, electricity and other belongings (e.g. television, bicycle, etc.). Principal component analysis (PCA) was used to measure the household wealth index [28,31,32]. The wealth index was then divided into quintiles (poorest/poorer/middle/richer/richest).

## Statistical analysis

Descriptive analysis was carried out to identify the socio-demographic characteristics of the respondents. The sample-weight of the NFHS-4 was used during descriptive analysis and it was reported in frequency and percentage. Then, bivariate analyses were carried out to determine the distribution of the independent variables across the BMI categories. The differences were identified using a chi-square ($\chi^2$) test. Given the hierarchical structure of the DHS data, multilevel ordered logistic regression was conducted to discern the factors associated with overweight and obesity [33–35]. At first, bivariate analyses were performed and those variables yielded a $p$-value <0.20. This was then placed in the multivariable model. This predefined $p$-value <0.20 was considered sufficient for preventing residual confounding in the multivariable model [36]. In the multivariable model, those variables yielding a p-value<0.05 were considered to be significant. To show the strength of association, both the unadjusted Crude Odds Ratio (COR) and the Adjusted Odds Ratio (AOR) were reported with a 95% confidence interval (CI). To resolve any multicollinearity among the covariates, Variance Inflation Factor (VIF) was tested to identify the presence of multicollinearity among the independent variables and a cut-off value of five was taken to indicate the presence of multicollinearity [37]. However, no significant multicollinearity was observed between any of the covariates. In the final ordered logistic regression analysis, the normal or underweight category (BMI <23.0 kg/m$^2$) was held as the reference group. All the statistical analyses were performed using Stata 14.0 (College Station, Texas, USA). The authors also followed the 'Strengthening the Reporting of Observational Studies in Epidemiology' (STROBE) statement in conducting this study and writing the manuscript (S1 File).

### Ethical approval

NFHS-4 protocol was approved by the institutional review board of IIPS and ICF International. Approval to use this dataset for secondary analysis was obtained from the DHS program on February 2019. Informed consent was obtained from each respondent before the interview.

## Findings

### Socio-demographic characteristics of the respondents

This study analysed data from 644,006 Indian women of reproductive age (15–49 years). The prevalence of overweight and obesity among the respondents was 22.7% (95% CI: 22.5%-22.9%) and 10.8% (95% CI: 10.6%-10.9%), respectively. The socio-demographic characteristics of the respondents along with the prevalence of the three categories of BMI across the covariates with the associated $\chi^2$ value are presented in Table 1. The majority of respondents were 35–49 years (36.7%), followed by 15–24 years (33.6%) and 25–34 years (29.7%). A majority of the respondents were from the rural area (65.5%). The highest proportion of the respondents (15.0%) hailed from Uttar Pradesh, followed by Maharashtra (9.6%). Nearly half of the respondents (47.3%) were educated up to the secondary level; however, more than a quarter of them (27.7%) received no formal education. Around one third of the respondents (30.2%) did not experience pregnancy, while cumulatively 69.8% of respondents had been pregnant at least once. Nearly three-fifths (61.4%) of the respondents reported to watching television almost every day at the time of the survey; however, 22.5% of the study participants reported that they did not watch television at all. The percentage of the respondents watching television less than once a week and at least once a week were 6.2% and 9.9%, respectively.

### Prevalence and distribution of overweight and obesity

The study found the BMI of the participants to vary across the covariates (Table 1). The prevalence of overweight and obesity increased with age and almost half of the women aged 35–49 years were either overweight (30.9%; 95% CI: 30.6%-31.2%) or obese (17.7%; 95% CI: 17.4%-18.0%). The prevalence was significantly higher among women living in urban areas compared to those living in rural areas (overweight: urban 28.7%; 95% CI: 28.3%-29.1% vs. rural 19.5%; 95% CI: 19.4%-19.7%; obesity: urban 17.8%; 95% CI: 17.4%-18.2% vs. rural 7.0%; 95% CI: 6.9%-7.2%). Women residing in Chandigarh, having higher educational attainment, belonging to the richest wealth quintile, being separated/divorced/widowed and having two children had a higher prevalence of overweight and obesity. The prevalence of overweight and obesity increased with increasing frequency of watching television and was the highest among the individuals who reported watching television almost every day. All the associated factors were statistically significant at a level of $p<0.001$.

A higher proportion of women in the urban area watched television almost every day than their rural counterparts (urban 81.7%; 95% CI: 81.2%-82.1% vs. rural 50.8%; 95% CI: 50.4%-51.1%, $p$-value$<0.001$). Similarly, 30.3% (95% CI: 30.0%-30.6%) of rural women did not watch television at all, which is much higher than their urban counterparts (7.6%; 95% CI: 7.3%-7.9%).

### Association between the frequency of watching television and overweight and obesity

The final model was adjusted for age, place and state of residence, wealth index, highest educational status, parity and number of household members. It was found that watching television was associated with overweight and obesity. Women who watched television less than once a

**Table 1. Weighted Prevalence of overweight and obesity in the sample population across the explanatory variables, NFHS 2015–16 (n = 644006).**

| Variable | n | (%) | BMI Status % (95% CI) | | |
|---|---|---|---|---|---|
| | | | BMI <23 (n = 428713) | 23≥ BMI <27.5 (n = 146041) | BMI ≥27.5 (n = 69251) |
| **Age Group (in years)** | | | | | |
| 15–24 | 216353 | (336) | 85.6 (85.4–85.9) | 11.3 (11.1–11.5) | 3.1 (3.0–3.3) |
| 25–34 | 191214 | (297) | 63.7 (63.3–64.1) | 25.5 (25.2–25.8) | 10.9 (10.6–11.1) |
| 35–49 | 236439 | (367) | 51.5 (51.1–51.9) | 30.9 (30.6–31.2) | 17.7 (17.4–18.0) |
| **Place of Residence** | | | | | |
| Rural | 421947 | (655) | 73.4 (73.2–73.7) | 19.5 (19.4–19.7) | 7.0 (6.9–7.2) |
| Urban | 222059 | (345) | 53.5 (53.0–54.1) | 28.7 (28.3–29.1) | 17.8 (17.4–18.2) |
| **State of Residence** | | | | | |
| Andaman and Nicobar Islands | 224 | (01) | 53.2 (50.0–56.4) | 29.6 (27.7–31.6) | 17.2 (14.8–20.0) |
| Andhra Pradesh | 27217 | (42) | 53.1 (51.7–54.6) | 27.5 (26.4–28.5) | 19.4 (18.4–20.5) |
| Arunachal Pradesh | 550 | (01) | 62.2 (60.9–63.5) | 30.2 (29.1–31.3) | 7.6 (6.7–8.3) |
| Assam | 16017 | (24) | 74.9 (74.0–75.8) | 19.7 (19.0–20.4) | 5.4 (5.1–5.9) |
| Bihar | 50454 | (78) | 79.0 (78.5–79.6) | 15.6 (15.2–16.1) | 5.4 (5.1–5.6) |
| Chandigarh | 491 | (01) | 44.0 (39.7–48.4) | 28.8 (25.3–32.7) | 27.2 (23.2–31.6) |
| Chhattisgarh | 15220 | (23) | 78.2 (77.3–79.0) | 16.4 (15.7–17.1) | 5.5 (5.1–6.0) |
| Dadra and Nagar Haveli | 158 | (01) | 71.1 (67.6–74.4) | 18.9 (16.5–21.6) | 9.9 (7.7–12.8) |
| Daman and Diu | 83 | (01) | 52.8 (48.3–57.3) | 30.8 (27.7–34.0) | 16.4 (13.8–19.5) |
| Goa | 833 | (01) | 49.1 (45.5–52.7) | 33.1 (30.5–35.8) | 17.8 (15.3–20.8) |
| Gujarat | 30137 | (46) | 64.5 (63.3–65.6) | 22.0 (21.2–22.9) | 13.5 (12.7–14.3) |
| Haryana | 14276 | (21) | 62.7 (61.7–63.8) | 26.9 (26.1–27.7) | 10.4 (9.7–11.0) |
| Himachal Pradesh | 3595 | (05) | 57.1 (55.6–58.6) | 27.8 (26.6.–29.0) | 15.1 (14.1–16.2) |
| Jammu and Kashmir | 6359 | (10) | 55.0 (53.8–56.2) | 29.3 (28.5–30.2) | 15.7 (14.8–16.6) |
| Jharkhand | 16183 | (25) | 81.1 (80.4–81.8) | 14.0 (13.5–14.6) | 4.9 (4.5–5.2) |
| Karnataka | 32555 | (51) | 63.5 (62.1–64.9) | 23.8 (22.9–24.7) | 12.7 (11.5–14.0) |
| Kerala | 18389 | (28) | 48.4 (47.2–49.5) | 36.2 (35.0–37.3) | 15.5 (14.5–16.5) |
| Lakshadweep | 41 | (01) | 44.5 (40.7–48.3) | 29.8 (27.3–32.4) | 25.8 (23.1–28,7) |
| Madhya Pradesh | 40326 | (63) | 76.1 (75.5–76.6) | 171 (16.7–17.6) | 6.8 (6.5–7.1) |
| Maharashtra | 61783 | (96) | 63.6 (62.5–64.7) | 23.6 (22.8–24.5) | 12.8 (12.1–13.6) |
| Manipur | 1135 | (02) | 55.3 (54.2–56.5) | 32.5 (31.5–33.4) | 12.2 (11.4–13.0) |
| Meghalaya | 1428 | (02) | 74.3 (72.8–75.7) | 21.3 (20.1–22.6) | 4.4 (3.8–5.1) |
| Mizoram | 542 | (01) | 61.1 (59.2–62.9) | 29.8 (28.1–31.6) | 9.2 (8.1–10.3) |
| Nagaland | 726 | (01) | 69.1 (67.8–70.3) | 24.2 (23.1–25.2) | 6.8 (6.1–7.5) |
| Delhi | 8153 | (13) | 51.4 (47.8–54.9) | 28.8 (26.2–31.5) | 19.8 (18.2–21.6) |
| Odisha | 23407 | (36) | 72.1 (71.4–72.9) | 19.9 (19.3–20.5) | 7.9 (7.5–8.4) |
| Puducherry | 756 | (01) | 46.8 (43.3–50.3) | 32.0 (28.7–35.5) | 21.3 (17.9–25.0) |
| Punjab | 14359 | (22) | 50.0 (48.9–51.1) | 32.6 (31.6–33.5) | 17.4 (16.6–18.3) |
| Rajasthan | 33791 | (53) | 74.7 (74.1–75.3) | 18.4 (17.9–18.9) | 7.0 (6.6–7.3) |
| Sikkim | 307 | (01) | 52.6 (50.9–54.3) | 34.3 (32.8–35.9) | 13.1 (12.0–14.4) |
| Tamil Nadu | 49064 | (76) | 53.1 (52.1–54.1) | 30.4 (29.7–31.2) | 16.5 (15.7–17.4) |
| Tripura | 2044 | (03) | 68.8 (66.9–70.6) | 24.8 (23.2–26.5) | 6.4 (5.7–7.2) |
| Uttar Pradesh | 96474 | (150) | 72.1 (71.7–72.5) | 19.7 (19.3–20.0) | 8.2 (8.0–8.5) |
| Uttarakhand | 5503 | (09) | 66.1 (64.8–67.4) | 23.2 (22.2–24.2) | 10.7 (9.9–11.5) |
| West Bengal | 51650 | (80) | 65.6 (64.4–66.7) | 25.2 (24.3–26.1) | 9.2 (8.6–10.0) |
| Telangana | 19776 | (31) | 58.6 (56.7–60.5) | 24.7 (23.2–26.4) | 16.6 (15.5–17.9) |
| **Highest Educational Status** | | | | | |

*(Continued)*

Table 1. (Continued)

| Variable | n | (%) | BMI Status % (95% CI) | | |
|---|---|---|---|---|---|
| | | | BMI <23 (n = 428713) | 23≥ BMI <27.5 (n = 146041) | BMI ≥27.5 (n = 69251) |
| No Formal Education | 178640 | (277) | 70.8 (70.5–71.2) | 21.2 (20.9–21.4) | 8.0 (7.8–8.2) |
| Primary | 80731 | (125) | 65.2 (64.6–65.7) | 23.6 (23.2–24.1) | 11.2 (10.8–11.6) |
| Secondary | 304321 | (473) | 66.4 (66.1–668) | 22.1 (21.8–2.2) | 11..5 (11.3–11.7) |
| Higher | 80314 | (125) | 59.1 (58.3–59.8) | 27.4 (26.8–2.8) | 136 (13.1–14.1) |
| **Wealth Index** | | | | | |
| Poorest | 112855 | (175) | 86.6 (86.4–86.9) | 11.2 (11.0–11.5) | 2.2 (2.1–2.3) |
| Poorer | 126139 | (196) | 78.0 (77.6–78.3) | 17.2 (16.9–17.5) | 4.8 (4.6–5.0) |
| Middle | 133091 | (207) | 68.1 (67.7–68.5) | 23.2 (22.8–23.5) | 8.8 (8.5–9.0) |
| Richer | 137204 | (213) | 56.9 (56.4–57.4) | 27.9 (27.5–28.3) | 15.2 (14.9–15.6) |
| Richest | 134717 | (209) | 47.4 (46.9–48.0) | 31.6 (31.1–32.1) | 21.0 (20.5–21.5) |
| **Marital Status** | | | | | |
| Single | 153999 | (239) | 86.3 (86.0–86.6) | 10.7 (10.5–11.0) | 3.0 (2.8–03.1) |
| Married | 461516 | (717) | 60.4 (60.1–60.7) | 26.4 (26.2–26.7) | 13.2 (13.0–13.4) |
| Separated/ Divorced/ Widowed | 28491 | (44) | 59.9 (59.0.-60.9) | 26.4 (25.6–27.2) | 13.7 (12.9–14.5) |
| **Parity** | | | | | |
| 0 | 194245 | (302) | 83.2 (82.9–83.5) | 12.6 (12.4–12.9) | 4.2 (4.0–4.4) |
| 1 | 83444 | (130) | 62.1 (61.5–62.7) | 25.9 (25.4–26.4) | 12.0 (11.6–12.4) |
| 2 | 163947 | (255) | 54.9 (54.4–55.3) | 29.3 (28.9–29.7) | 15.9 (15.5–16.2) |
| 3 | 101464 | (158) | 59.0 (58.5–59.4) | 27.2 (26.8–27.6) | 13.9 (13.5–14.3) |
| 3+ | 100906 | (157) | 64.9 (64.5–65.4) | 24.1 (23.8–24.5) | 11.0 (10.7–11.3) |
| **Number of Household Members** | | | | | |
| ≤5 | 370677 | (576) | 63.2 (62.9–63.5) | 24.6 (24.4–24.8) | 123 (12.0–12.5) |
| >5 | 273329 | (424) | 71.2 (70.9–71.5) | 20.1 (19.9–20.3) | 87 (8.5–8.9) |
| **Frequency of Watching Television** | | | | | |
| Not at all | 144768 | (225) | 80.1 (79.8–80.5) | 15.3 (15.0–15.5) | 4.6 (4.4–4.8) |
| Less than once a week | 39761 | (62) | 76.3 (75.7–76.9) | 17.6 (17.0–18.1) | 6.1 (5.8–6.5) |
| At least once a week | 63861 | (99) | 71.1 (70.6–71.6) | 20.6 (20.2–21.1) | 8.3 (7.9–8.6) |
| Almost every day | 395616 | (614) | 59.9 (59.6–60.2) | 26.2 (26.0–26.5) | 13.9 (13.6–14.1) |

NFHS: National Family Health Survey

CI: Confidence Interval.

week were only 1.05 times more likely to be overweight and obese than the women who never watched television (AOR: 1.05, 95% CI: 1.02–1.08). Comparatively, women watching television at least once a week and almost every day had 10% (AOR: 1.10, 95% CI: 1.08–1.13) and 24% (AOR: 1.24, 95% CI: 1.21–1.26) increased odds of being overweight and obese, respectively, compared to their counterparts who never watched television (Table 2). All the associated were statistically significant at a level of $p<0.001$.

Upon stratification by urban and rural residence, overweight and obesity showed significant association with the frequency of viewing television in both urban and rural areas. In urban areas, watching television almost every day increased the odds of overweight and obesity 1.15 times compared to not watching television at all (AOR: 1.15, 95% CI: 1.10–1.20; $p$-value<0.001). In rural areas, women watching television less than once a week, at least once a week and almost every day had 6% (AOR: 1.06, 95% CI: 1.03–1.09; $p$-value<0.001), 11% (AOR: 1.11, 95% CI: 1.08–1.14; $p$-value<0.001) and 26% (AOR: 1.26, 95% CI: 1.23–1.29; $p$-

**Table 2. Association between the frequency of viewing Television and overweight and obesity among reproductive age women of India, NFHS 2015–16.**

| Frequency of viewing Television | COR (95% CI) | AOR (95% CI) |
|---|---|---|
| **Total:** | | |
| Not at all | Ref | Ref |
| Less than once a week | 1.18 (1.15–1.22) | 1.05 (1.02–1.08) |
| At least once a week | 1.47 (1.44–1.50) | 1.10 (1.08–1.13) |
| Almost every day | 1.93 (1.89–1.96) | 1.24 (1.21–1.26) |
| **In Urban Area:** | | |
| Not at all | Ref | Ref |
| Less than once a week | 1.10 (1.04–1.17) | 1.00 (0.94–1.07) |
| At least once a week | 1.30 (1.24–1.36) | 1.05 (1.00–1.10) |
| Almost every day | 1.52 (1.46–1.58) | 1.15 (1.10–1.20) |
| **In Rural Area:** | | |
| Not at all | Ref | Ref |
| Less than once a week | 1.15 (1.12–1.19) | 1.06 (1.03–1.09) |
| At least once a week | 1.41 (1.37–1.44) | 1.11 (1.08–1.14) |
| Almost every day | 1.84 (1.81–1.88) | 1.26(1.23–1.29) |

NFHS: National Family Health Survey

COR: Crude Odds Ratio

CI: Confidence Interval

AOR: Adjusted Odds Ratio

Results are based on ordered logistic regression and adjusted for age, place of residence, province of residence, ecological zone of residence, highest educational status, wealth index, parity and number of household members.

BMI <23 group was considered as the reference group.

value<0.001) increased odds of being overweight and obese compared to women not watching television at all (Table 2). The final logistic regression models are shown in Tables A-C in S2 File. In the final model, interaction effect between the frequency of television viewing and the place of residence was significant (S2 File).

## Discussion

To the best of our knowledge, this was the first study that investigated the association of frequency of television watching with overweight and obesity among reproductive age Indian women. After analysing the data from a weighted sample of over half a million participants, this study found that the likelihood of being overweight and obese significantly increased with the frequency of watching television, particularly in the rural area. The increasing burden of overweight and obesity, especially among women, is contributing to the NCD epidemic in India [38]. High BMI among women is also associated with pregnancy complications, caesarean delivery and giving birth to a preterm baby [39]. As previously discussed, the adverse effect of overweight and obesity on women of reproductive age, including infertility and miscarriage, is of great importance to the public health field [23–24]. Therefore, it is important to study the associated factors of overweight and obesity among this target group.

The current study found that around one third of Indian women of reproductive age were either overweight or obese (33.5%). A similar prevalence of overweight and obesity in this target age group was found in other neighbouring countries of India (i.e. Bangladesh, Pakistan and Myanmar) using the Asian cut-off and nationally representative data. The prevalence was found to be 36% in Bangladesh, 38.7% in Myanmar and 39% in Pakistan [27,40,41]. The

prevalence of overweight and obesity was higher among women from the older age group (35–49 years), which is consistent with other studies [13,27]. This may be partly because people in general are less capable of engaging in physical activity as they age but do not adjust their dietary and lifestyle habits accordingly [13]. The prevalence of overweight and obesity also increased with higher educational attainment and an increased wealth index, which is also consistent with the literature [27,42,43]. The individuals with higher educational attainment and from the richest wealth quintiles in the low- and middle-income countries are involved in less manual work, which leads to gain excessive weight gain [42].

The frequency of watching television was significantly higher among the urban women than the rural women. This observation has also been made in Bangladesh, Myanmar and Nepal [15,27,43]. This is most probably due to the increased accessibility for urban residents in terms of a variety of amenities including, stable electric supply and satellites channels [15,27].

This study found that watching television was significantly associated with overweight and obesity in women of reproductive age, which is consistent with studies done in Bangladesh and Myanmar [15,27]. A similar association was observed among children and adolescents [16,25,44,45] and adult males [26]. It was also observed that in urban areas, watching television almost every day was only found to be positively associated with overweight and obesity. But in the rural area, any frequency of watching television (less than once a week, at least once a week and almost every day) was found to be significant. This may be because of the obesogenic environment that exists in urban areas; i.e. the increased availability and accessibility of fast foods [46]. This is further exacerbated by the promotion of obesogenic foods through television advertisement, which has been shown to have an impact on a viewer's dietary habits [47]. Urban residents in India are also less physically active and are prone to a sedentary lifestyle [48]. All these established risk factors for overweight and obesity may have overridden the effect of watching television [27]. This may explain why the most extreme option, watching television almost every day, was the only one that showed a significant association only. In the rural area, the prevalence of overweight and obesity is lower due to the increased level of physical activity, which is partially due to the less developed transportation system and increased prevalence of manual work as well as the reduced availability and thus consumption of junk food [15,27].

The Government of India has formulated a 'National Action Plan and Monitoring Framework' for the prevention and control of NCDs. The action plan aims for a 10% increase in the physical activity level by 2025 [45]. Public health programs aiming to prevent NCDs should endorse physical activity as a non-pharmaceutical preventive measure as well as the harmful effects of extended time watching television especially in lieu of more physically enriching leisure activities. Further research is needed to find the effects of television watching on children, adolescents and adult males.

## Strengths and limitations

This study utilized a nationally representative sample collected from every state and union territory of India and from both urban and rural areas, so the findings of the study are generalizable in the context of India. Moreover, as a result of the utilization of a validated questionnaire and calibrated measuring tools by the NFHS-4, the possibility of measurement errors is very unlikely compared to any other cross-sectional studies in India. Unlike previous studies based on DHS data, which measured the frequency of television watching in weeks, the NFHS-4 measured the frequency one category in days, which gives a more precise estimate [18,24].

The limitations of the study also warrant discussion. First, this is a cross-sectional study, so the temporal relationship between the outcome variable and the explanatory variable could

not be established. Second, information on physical activity levels, food habits of the participants and the frequency of using other devices, such as computers or mobiles, specifically as a means of entertainment, were not collected. As such, those variables could not be included in the analyses.

## Conclusions

The findings from this study suggest that watching television is positively associated with overweight and obesity in Indian women of reproductive age. Considering the consequences of overweight and obesity, public health promotion programs in India must create awareness about the harmful effects of sedentary lifestyles due to watching television frequently. Further research should investigate the condition in the general population.

## Supporting information

**S1 File. STROBE Checklist.**
(DOCX)

**S2 File. Supplementary Tables.**
(DOCX)

## Acknowledgments

We are grateful to the DHS program for providing access to the dataset.

## Author Contributions

**Conceptualization:** Rajat Das Gupta, Shams Shabab Haider, Ipsita Sutradhar.

**Data curation:** Rajat Das Gupta, Shams Shabab Haider.

**Formal analysis:** Rajat Das Gupta, Mohammad Rashidul Hashan, Ibrahim Hossain Sajal, Mehedi Hasan, Mohammad Rifat Haider, Malabika Sarker.

**Investigation:** Rajat Das Gupta, Shams Shabab Haider, Ipsita Sutradhar, Mohammad Rashidul Hashan, Mohammad Rifat Haider.

**Methodology:** Rajat Das Gupta, Shams Shabab Haider, Ipsita Sutradhar, Mohammad Rashidul Hashan, Ibrahim Hossain Sajal, Mehedi Hasan, Mohammad Rifat Haider, Malabika Sarker.

**Project administration:** Rajat Das Gupta.

**Resources:** Rajat Das Gupta.

**Software:** Rajat Das Gupta.

**Supervision:** Mohammad Rifat Haider, Malabika Sarker.

**Validation:** Rajat Das Gupta.

**Visualization:** Rajat Das Gupta, Mohammad Rashidul Hashan, Ibrahim Hossain Sajal, Mehedi Hasan.

**Writing – original draft:** Rajat Das Gupta, Shams Shabab Haider, Ipsita Sutradhar.

**Writing – review & editing:** Rajat Das Gupta, Shams Shabab Haider, Ipsita Sutradhar, Mohammad Rashidul Hashan, Ibrahim Hossain Sajal, Mehedi Hasan, Mohammad Rifat Haider, Malabika Sarker.

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
