## [Decision Letter · Decision Letter 0]

12 Jul 2019

PONE-D-19-16263

Association of frequency of television watching with overweight and obesity among women of reproductive age in India: Evidence from a nationally representative study

PLOS ONE

Dear Dr. Das Gupta,

Thank you for submitting your manuscript to PLOS ONE. After careful consideration, we feel that it has merit but does not fully meet PLOS ONE’s publication criteria as it currently stands. Therefore, we invite you to submit a revised version of the manuscript that addresses the points raised during the review process.

The scientific rationale should be reviewed and strengthened. Reviewer 1 and myself take issue with p-values and CI. Table 1 simply does not need any p-values, especially not in the form proposed (***). Accordingly, the text needs changes, you can simply state that the differences were all stat.significant at a level of p<0.001).

For the AOR, where you show CI, please refrain from p-values, and thus delete all the asterisks from table 2.

Please also revise the discussion to reflect the rationale and the importance of the findings in this context.

We would appreciate receiving your revised manuscript by Aug 26 2019 11:59PM. To enhance the reproducibility of your results, we recommend that if applicable you deposit your laboratory protocols in protocols.io, where a protocol can be assigned its own identifier (DOI) such that it can be cited independently in the future. For instructions see: http://journals.plos.org/plosone/s/submission-guidelines#loc-laboratory-protocols

We look forward to receiving your revised manuscript.

Kind regards,

Hajo Zeeb

Academic Editor

PLOS ONE

Journal Requirements:

Reviewers' comments:

Reviewer's Responses to Questions

**Comments to the Author**

1. Is the manuscript technically sound, and do the data support the conclusions?

Reviewer #1: Yes

Reviewer #2: Yes

2. Has the statistical analysis been performed appropriately and rigorously? 

Reviewer #1: No

Reviewer #2: Yes

3. Have the authors made all data underlying the findings in their manuscript fully available?

Reviewer #1: Yes

Reviewer #2: Yes

4. Is the manuscript presented in an intelligible fashion and written in standard English?

Reviewer #1: Yes

Reviewer #2: Yes

5. Review Comments to the Author

Reviewer #1: Journal

Plos One

Manuscript ID

PONE-D-19-16263

Title

Association of frequency of television watching with overweight and obesity among women of reproductive age in India: Evidence from a nationally representative study

Comments

Thank you for the opportunity of reading this paper. This is a simple paper, but it is related with an important topic.

I enjoy reading it, but I have some concerns and comments that I address to the authors.

Introduction

I start reading the paper with great expectation. The rationale was OK, but it was not explore why is important to study the relationship between overweight/obesity and sedentary behaviour among women of reproductive age. Are these women at risk of being obese? Well, the authors did not present any indication of that.

Page 7, line 115. I am not aware what is the recommended age for being pregnant. Why authors used women aged 15-49 as sample?

Page 9, lines 153-155. How was used the PCA to measure the household wealth index? Can the authors describe the results of the PCA?

Page 15, lines 209-218 (table 1). Looking to the table 1, we can see that there are significant differences among the categories of all variable. If fact this was expected because the sample is huge. Therefore, p-value in this cases is not so important. Can the authors provide the 95% CI for the % of BMI status?

I know that it is a boring work for the authors to provide the 95% CI for all numbers, but it seems to be important.

Page 14, lines 220-223. If the authors want to present a figure, it would be more important to use a figure focus on study aim. For instance, it could be a figure showing the relationship of watch television and obesity.

Page 16, table 2. Asterisks should be put after the CI. For example: 1.05 (1.02-1.08)***, instead of 1.05***(1.02-1.08).

Page 18, lines 295-296. Why women who watch TV in the rural areas tend to adopt a sedentary lifestyle? I am asking that because women can watch TV at night, after a workday and it might not be related to a sedentary lifestyle.

Page 18, line 298. Delete the word "ect".

Page 18, lines 303-305. A dose-response relationship cannot be observed in a cross sectional study. This assumption is exaggerated.

The discussion has the same weakness as the introduction section. Authors do not explain why it is important to study obesity among women of reproductive age. Without this explanation this is only on more study that tried to explained the relationship between obesity and time spend watching TV.

Reviewer #2: This study investigated the Association of frequency of television watching with overweight and obesity among women of reproductive age in India. The method used was the Strengthening the Reporting of Observational Studies in Epidemiology (STROBE) checklist for cross-sectional studies. The paper referred to relevant studies throughout the introduction and discussion sections, citing relevant populations and recent findings. The authors raised pertinent issues around the association of obesity and diferents factors of risk. The suggestions around prevention programs. The results were reported clearly and the tables were well presented.

6. PLOS authors have the option to publish the peer review history of their article (what does this mean?). If published, this will include your full peer review and any attached files.

Reviewer #1: Yes: Adilson Marques

Reviewer #2: Yes: David Rincon-Pabon

---

## [Author Response · Author response to Decision Letter 0]

16 Jul 2019

Title of the Manuscript: Association of frequency of television watching with overweight and obesity among women of reproductive age in India: Evidence from a nationally representative study

Reference Number: PONE-D-19-16263

# Editor:

Dear Dr. Das Gupta,

Thank you for submitting your manuscript to PLOS ONE. After careful consideration, we feel that it has merit but does not fully meet PLOS ONE’s publication criteria as it currently stands. Therefore, we invite you to submit a revised version of the manuscript that addresses the points raised during the review process.

The scientific rationale should be reviewed and strengthened. 

Response: Thank you for this important suggestion. We have now revised and reorganized the introduction section as following: “Women of reproductive age that settle into a sedentary lifestyle are at risk of being overweight/obese [19]. For women of reproductive age, overweight and obesity is an established risk factor for several medical complications, including pre-eclampsia, eclampsia and gestational diabetes mellitus (GDM) [20,21]. Overweight and obese women having GDM also tend to develop diabetes mellitus during the post-partum period [22]. Overweight and obesity is attributable to an increased risk of infertility and miscarriage[23]. Living a sedentary lifestyle is also an independent risk factor for infertility in women [24].”

Reviewer 1 and myself take issue with p-values and CI. Table 1 simply does not need any p-values, especially not in the form proposed (***). Accordingly, the text needs changes, you can simply state that the differences were all stat.significant at a level of p<0.001).

Response: Thank you! We have revised the table and the text accordingly. 

For the AOR, where you show CI, please refrain from p-values, and thus delete all the asterisks from table 2.

Response: Thank you! We have revised the table and the text accordingly.

Please also revise the discussion to reflect the rationale and the importance of the findings in this context.

Response: Thanks for this important suggestion! We have revised the discussion section and have added the following portion explaining why it is important to study obesity among women of reproductive age: “The increasing burden of overweight and obesity, especially among women, is contributing to the NCD epidemic in India [38]. High BMI among women is also associated with complicated pregnancy, caesarean delivery and giving birth to preterm baby [39]. As previously discussed, the adverse effect of overweight and obesity on women of reproductive age including infertility and miscarriage has also public health importance [23-24]. Therefore it is important to study the associated factors of overweight and obesity among this target group.” 

Reviewer #1:

Title

Association of frequency of television watching with overweight and obesity among women of reproductive age in India: Evidence from a nationally representative study

Comments

Thank you for the opportunity of reading this paper. This is a simple paper, but it is related with an important topic. I enjoy reading it, but I have some concerns and comments that I address to the authors.

Introduction

I start reading the paper with great expectation. The rationale was OK, but it was not explore why is important to study the relationship between overweight/obesity and sedentary behaviour among women of reproductive age. Are these women at risk of being obese? Well, the authors did not present any indication of that.

Response: Thank you for this important suggestion. We have now revised and reorganized the introduction section as following: “Women of reproductive age that settle into a sedentary lifestyle are at risk of being overweight/obese [19]. For women of reproductive age, overweight and obesity is an established risk factor for several medical complications, including pre-eclampsia, eclampsia and gestational diabetes mellitus (GDM) [20,21]. Overweight and obese women having GDM also tend to develop diabetes mellitus during the post-partum period [22]. Overweight and obesity is attributable to an increased risk of infertility and miscarriage [23]. Living a sedentary lifestyle is also an independent risk factor for infertility in women [24].”

Page 7, line 115. I am not aware what is the recommended age for being pregnant. Why authors used women aged 15-49 as sample?

Response: Thank you Honorable reviewer for this comment. We considered 15-49 years as reproductive age group as per the standard world health organization definition (https://apps.who.int/iris/bitstream/handle/10665/43185/924156315X_eng.pdf;jsessionid=E708E1B5028C684F6FB8778623B8828C?sequence=1). The DHS program usually collects data among this age group. NFHS-4 collected data between 15-54 years age group. As previous studies utilizing the DHS data analyzed samples between this age group (https://bmjopen.bmj.com/content/9/3/e024680.info, https://bmjopen.bmj.com/content/7/1/e014399) , we decided to do the same for harmonization of the findings across countries of South Asia. Also we mentioned the consequences of overweight and obesity in this age group in the introduction section: “Women of reproductive age that settle into a sedentary lifestyle are at risk of being overweight/obese [19]. For women of reproductive age, overweight and obesity is an established risk factor for several medical complications, including pre-eclampsia, eclampsia and gestational diabetes mellitus (GDM) [20,21]. Overweight and obese women having GDM also tend to develop diabetes mellitus during the post-partum period [22]. Overweight and obesity is attributable to an increased risk of infertility and miscarriage [23]. Living a sedentary lifestyle is also an independent risk factor for infertility in women [24].”

Page 9, lines 153-155. How was used the PCA to measure the household wealth index? Can the authors describe the results of the PCA?

Response: Thanks! The PCA was done by the National Family Health Survey (NFHS-4) authority and the variable was incorporated within the dataset provided by DHS. The results of the PCA was not described in the NFHS-4 final report. 

Page 15, lines 209-218 (table 1). Looking to the table 1, we can see that there are significant differences among the categories of all variable. If fact this was expected because the sample is huge. Therefore, p-value in this cases is not so important. Can the authors provide the 95% CI for the % of BMI status? I know that it is a boring work for the authors to provide the 95% CI for all numbers, but it seems to be important.

Response: Thanks for this important suggestion! We have removed the p-value and added the 95% CI for the % of BMI status in Table 1.

Page 14, lines 220-223. If the authors want to present a figure, it would be more important to use a figure focus on study aim. For instance, it could be a figure showing the relationship of watch television and obesity.

Response: Thanks! We have removed Figure 1. As we have shown the relationship of watch television and obesity in Table 1, we did not include it as a figure again.

Page 16, table 2. Asterisks should be put after the CI. For example: 1.05 (1.02-1.08)***, instead of 1.05***(1.02-1.08).

Response: Thanks! As per the suggestion of the honorable editor, we removed the asterisk from table 2. 

Page 18, lines 295-296. Why women who watch TV in the rural areas tend to adopt a sedentary lifestyle? I am asking that because women can watch TV at night, after a workday and it might not be related to a sedentary lifestyle.

Response: Thanks for this important observation. We have removed this sentence.

Page 18, line 298. Delete the word "ect".

Response: Thanks! We have deleted the word as per suggestion. 

Page 18, lines 303-305. A dose-response relationship cannot be observed in a cross sectional study. This assumption is exaggerated.

Response: Thanks! We have removed the sentence.

The discussion has the same weakness as the introduction section. Authors do not explain why it is important to study obesity among women of reproductive age. Without this explanation this is only on more study that tried to explained the relationship between obesity and time spend watching TV.

Response: Thank you for this important suggestion. We have now revised and reorganized the introduction section as following: “The increasing burden of overweight and obesity, especially among women, is contributing to the NCD epidemic in India [38]. High BMI among women is also associated with pregnancy complications, caesarean delivery and giving birth to a preterm baby [39]. As previously discussed, the adverse effect of overweight and obesity on women of reproductive age, including infertility and miscarriage, is of great importance to the public health field [23-24].Therefore, it is important to study the associated factors of overweight and obesity among this target group.”

Reviewer #2

 This study investigated the Association of frequency of television watching with overweight and obesity among women of reproductive age in India. The method used was the Strengthening the Reporting of Observational Studies in Epidemiology (STROBE) checklist for cross-sectional studies. The paper referred to relevant studies throughout the introduction and discussion sections, citing relevant populations and recent findings. The authors raised pertinent issues around the association of obesity and diferents factors of risk. The suggestions around prevention programs. The results were reported clearly and the tables were well presented.

Response: Thank you very much for your valuable comments.

---

## [Editor Report · Decision Letter 1]

15 Aug 2019

Association of frequency of television watching with overweight and obesity among women of reproductive age in India: Evidence from a nationally representative study

PONE-D-19-16263R1

Dear Dr. Das Gupta,

We are pleased to inform you that your manuscript has been judged scientifically suitable for publication and will be formally accepted for publication once it complies with all outstanding technical requirements.

With kind regards,

Hajo Zeeb

Academic Editor

PLOS ONE
---

## [Editor Report · Acceptance letter]

21 Aug 2019

PONE-D-19-16263R1 

Association of frequency of television watching with overweight and obesity among women of reproductive age in India: evidence from a nationally representative study 

Dear Dr. Das Gupta:

I am pleased to inform you that your manuscript has been deemed suitable for publication in PLOS ONE. Congratulations! Your manuscript is now with our production department. 

With kind regards,

on behalf of

Prof. Hajo Zeeb 

Academic Editor

PLOS ONE